# Heavy-Metal Contents and the Impact of Roasting on Polyphenols, Caffeine, and Acrylamide in Specialty Coffee Beans

**DOI:** 10.3390/foods10061310

**Published:** 2021-06-07

**Authors:** Matúš Várady, Sylwester Ślusarczyk, Jana Boržíkova, Katarína Hanková, Michaela Vieriková, Slavomír Marcinčák, Peter Popelka

**Affiliations:** 1Department of Food Hygiene, Technology and Safety, University of Veterinary Medicine and Pharmacy, Komenského 73, 041 81 Košice, Slovakia; slavomir.marcincak@uvlf.sk; 2Department of Pharmaceutical Biology and Botany, Wroclaw Medical University, Borowska 211, 50-556 Wroclaw, Poland; sylwester.slusarczyk@umed.wroc.pl; 3State Veterinary and Food Institute Dolný Kubin, Hlinkova 619, 043 65 Košice, Slovakia; borzikova@svu-ke.sk; 4State Veterinary and Food Institute Dolný Kubin, Jánošková 1611, 026 01 Dolný Kubín, Slovakia; katarina.hankova@svpu.sk (K.H.); miska.vierikova@svpu.sk (M.V.)

**Keywords:** specialty coffee, roasting, chlorogenic acids, caffeine, acrylamide, heavy metals, ultra-high-resolution mass spectrometry, tandem mass spectroscopy

## Abstract

The aim of this study was to determine the effect of roasting on the contents of polyphenols (PPH), acrylamide (AA), and caffeine (CAF) and to analyze heavy metals in specialty coffee beans from Colombia (COL) and Nicaragua (NIC). Samples of NIC were naturally processed and COL was fermented anaerobically. Green beans from COL (COL-GR) and NIC (NIC-GR) were roasted at two levels, light roasting (COL-LIGHT and NIC-LIGHT) and darker roasting (COL-DARK and NIC-DARK), at final temperatures of 210 °C (10 min) and 215 °C (12 min), respectively. Quantitative analyses of PPH identified caffeoylquinic acids (CQA), feruloylquinic acids, and dicaffeoylquinic acids. Isomer 5-CQA was present at the highest levels and reached 60.8 and 57.7% in COL-GR and NIC-GR, 23.4 and 29.3% in COL-LIGHT and NIC-LIGHT, and 18 and 24.2% in COL-DARK and NIC-DARK, respectively, of the total PPH. The total PPH contents were highest in COL-GR (59.76 mg/g dry matter, DM). Roasting affected the contents of PPH, CAF, and AA (*p* < 0.001, *p* < 0.011 and *p* < 0.001, respectively). Nickel and cadmium contents were significantly higher in the COL-GR than in the NIC-GR beans. Darker roasting decreased AA content, but light roasting maintained similar amounts of CAF and total PPH.

## 1. Introduction

The antioxidant activity of coffee beans depends on the characteristics of phenolic compounds, especially chlorogenic acids (CGA) and caffeine (CAF). Both CGA and CAF are important for flavor formation and the healthy effect of coffee brews and their extracts because they reduce oxidative stress and protect the antioxidant system [1,2]. CGA possesses a wide range of potential health benefits, which have been ascribed to the acyl-quinic acids in the brew and are associated with coffee consumption, including reduced incidences of several chronic and degenerative diseases such as cancer, cardiovascular disorders, diabetes, and Parkinson’s disease [3]. Specialty coffee is the term for coffee that has a standardized process of production, from choosing criteria for coffee plantations to coffee brews that pass preliminary tests of grading and cupping. Only then can specialty coffees be served to the client [4,5]. One of the most important criteria is to achieve a cupping score of 80 points or above on the 100 point scale [4]. Farmers who process specialty coffee use the new types of coffee processing (anaerobic fermentation, carbonic macerations, thermal shock, etc.) to maintain the quality of coffee. Roasting is the most important operation responsible for the main chemical, physical, and organoleptic characteristics of the final coffee product [6]. When coffee is roasted at high temperatures, potentially toxic substances such as acrylamide (AA) are formed, mainly due to Maillard reactions between asparagine and reduced sugars such as glucose and fructose [6,7]. The development of AA depends on factors such as time and temperature, the degree of roasting, and the origin of the coffee [8,9]. AA develops early during roasting, reaching a maximum level within a few minutes, and then begins to decrease [10].

CAF is the most routinely ingested bioactive substance throughout the world. It is a natural methylxanthine alkaloid found in more than 60 plants, including coffee beans, tea leaves, cola nuts, and cocoa pods [11]. The quantity of CAF is greatly affected by the roasting level and decreases with the intensity of roasting [12,13].

Heavy metals are evaluated in coffees because they can be absorbed and stored by coffee plants in the roots, shoots, or grains, but the contents are usually much higher in the vegetative components than the grains [14]. Heavy metals are sources of contamination for humans because they accumulate in the food chain [15]. The bioavailability and occurrence of heavy metals, however, are influenced by other factors such as soil pH, cation-exchange capacity, organic-matter content, soil texture, and interactions among elements [16].

The effect of roast degree on the contents of AA, CAF, and polyphenols (PPH) has already been studied extensively [17,18,19]. However, the influence of coffee type and roasting on the contents of total PPH, CAF, and AA has not yet been sufficiently studied in specialty coffees. We hypothesized that heavy-metal contents would vary substantially among green specialty coffee beans from different areas. Our goals were therefore to (1) identify the main bioactive compounds in specialty coffee beans (Colombian (COL) and Nicaraguan (NIC)); (2) determine the effects of coffee type and roasting on the contents of total PPH, CAF, and AA; and (3) determine the contents of heavy metals in green specialty coffee beans.

## 2. Materials and Methods

### 2.1. Coffee Samples

Samples of 100% COL and NIC *Coffea arabica* beans (Project FOX Roastery, Prague, Czech Republic) were used in our experiment. The COL coffee was from the Finca La Cabana farm and was processed by anaerobic fermentation. The coffee seeds were fermented in plastic barrels sealed with one-way valves to prevent oxygen from entering but allowing oxygen to escape as the coffee ferments. Attaining an oxygen-free environment took about 25 h. The coffees were fermented anaerobically for 200 h and then sun-dried. Parchment coffees were stored for 2–3 months to stabilize water activity and humidity. The NIC coffee was from the Finca Buena Vista farm and was processed using the natural method. It is a unique variety of Sarchimor, which is a coffee hybrid.

### 2.2. Roasting

The roasting of the green COL and NIC beans (COL-GR and NIC-GR, respectively) was done based on the degree of roasting. The light roasting (COL-LIGHT and NIC-LIGHT) was at a final temperature of 210 °C for 10 min, and the development time was 30 s. The darker roasting (COL-DARK and NIC-DARK, respectively) was at a final temperature of 215 °C for 12 min, and the development time was 2 min and 30 s. The coffee was roasted in batch sizes of 0.25 kg in a Probatone 5 gas roaster (Probat, Emmerich am Rhein, Germany). The temperature was increased over time and depended on the amount of gas provided. The temperature was measured with a probe inside the drum of the roaster.

### 2.3. Analysis of PPH

Coffee samples were ground to fine powders using an FW 177 Herbal Medicine Disintegrator (Chincan, Tianjin, China) at 24, 000 rpm, and each 100 mg was extracted three times with 80% MeOH at 40 °C for 35 min in an ultrasonic bath. Particulate extracts were combined and evaporated to dryness and were then dissolved in 2 mL of Milli-Q water (acidified with 0.2% formic acid (FA)) and purified by solid-phase extraction (SPE) using a 60-mg Oasis HLB 3 cc Vac Cartridge (Waters Corp., Milford, CT, USA). The cartridges were washed with 0.5% methanol to remove carbohydrates and then washed with 80% methanol to elute the polyphenolic fraction. The polyphenolic fraction was re-evaporated to dryness and dissolved in 1 mL of 80% methanol (acidified with 0.2% FA). The sample was then centrifuged at 23, 000× *g* for 5 min and diluted five-fold with Milli-Q water before spectrometric analysis. All analyses were performed in triplicate for two independent samples and stored at −20 °C before analysis.

### 2.4. Ultra-High-Resolution Mass Spectrometry (UHRMS)

The particulate compounds from the 80% MeOH SPE fraction containing PPH were analyzed using UHRMS on a Dionex UltiMate 3000RS system (Thermo Scientific, Darmstadt, Germany) with a charged aerosol detector connected to a compact QTOF mass spectrometer (Bruker Daltonik GmbH, Bremen, Germany). The coffee samples were chromatographically separated using a BEH C18 column (2.1 × 100 mm, 1.7 μM, Waters) with mobile phase A consisting of 0.1% (*v*/*v*) FA in water and mobile phase B consisting of 0.1% FA (*v*/*v*) in acetonitrile. The conditions of the initial mobile phase were 7% B and 93% A held for 1 min; then, from 1 to 20 min, it was ramped concavely from 7% to 50% phase B in phase A to separate the phenolic compounds with a short 0.3 min calibration segment from 0 to 0.5 min. The flow rate was 0.3 mL/min, and the column was held at 25 °C. Spectra were acquired in negative-ion mode over a mass range from *m*/*z* 100 to 1500 with a frequency of 5 Hz. The operating parameters of the electrospray ion source were: capillary voltage, 3 kV; dry gas flow, 6 L/min; dry gas temperature, 200 °C; nebulizer pressure, 0.7 bar; collision radio frequency, 6000 V; transfer time, 70.0 μs; and prepulse storage, 7.0 μs. Ultrapure nitrogen was used as the drying and nebulizer gas, and argon (Ar) was used as the collision gas. The collision energy was set automatically from 15 to 75 eV depending on the *m/z* of the fragmented ion. The data were calibrated internally with sodium formate introduced to the ion source at the beginning of each separation via a 20 μL loop. Spectral data were acquired and processed using Bruker DataAnalysis 4.3 software. The concentrations of the phenolic compounds were calculated as equivalents of chlorogenic acid. Calibration curves were constructed based on five concentration points (from 0.23 to 0.000718 mg/mL) using Bruker QuantAnalysis 4.3 software. All analyses were performed in triplicate.

### 2.5. Analysis of CAF

The samples of ground coffee were homogenized with the addition of Carez I (K_4_[Fe (CN)_6_]3H_2_O) and Carez II (ZnSO_4_7H_2_O) solutions and extracted at high temperature. High-Performance Liquid Chromatography HPLC analyses were performed on an AGILENT 1200 chromatograph (Agilent Technologies, Santa Clara, CA, USA) equipped with a DAD detector (212 nm). The samples were chromatographically separated using an LC 18 column (4 × 125 mm, 5 μM) (Merck KGaA, Darmstadt, Germany) with mobile phase A consisting of an acetate buffer and mobile phase B consisting of acetonitrile, with isocratic elution [20]. The CAF peak was detected at a wavelength of 212 nm.

### 2.6. Analysis of AA

The samples of ground coffee were homogenized with the addition of the Carez I and Carez II solutions. The samples were extracted using acetonitrile and hexane, SPE, glass column packed with Hydromatrix, and evaporated on a rotary evaporator. The AA contents of the roasted samples were determined using a Dionex Ultimate 3000 chromatograph (Thermo Scientific, Waltham, MA, USA). The compounds were separated using a C18 Waters Acquity UPLC BEH chromatographic column (2.1 × 100 mm, 1.7 μM,). The mobile phase was acetonitrile and water (FA as an additive), with gradient elution. AA was detected using an ABSCIEX QTRAP 5500 system (Sciex, Framingham, MA, USA) [21,22].

### 2.7. Analysis of Heavy Metals

The coffee samples were ground to a standard grind and homogenized, weighed (0.3 g), and transferred to Merck Teflon beakers (Merck KGaA, Darmstadt, Germany) for mineralization. Five milliliters of 65% nitric acid and 1 mL of hydrogen peroxide were added. The Merck Teflon beakers were sealed and mineralized for 45 min. The sample solutions were then quantitatively transferred to 25 mL volumetric flasks, and deionized water was added to bring the volume to 25 mL for analysis. The samples were digested in a Berghof MWS-2 microwave device (Berghof Automation GmbH, Eningen unter Achalm, Germany) for 45 min at temperatures ranging from 100 to 190 °C. Heavy metals (copper, lead, chrome, cadmium (Cd), nickel (Ni), aluminum, and mercury) were identified using the multi-element technique of inductively coupled plasma mass spectrometry (Agilent ICP-MS 7900, Santa Clara, CA, USA). The ICP-MS conditions were: RF power, 1370 W; RF matching, 1.58 V; plasma gas (Ar) flow, 15 L/min; sampling depth, 7.9 mm; carrier gas (Ar) flow, 1.22 L/min; nebulizer pump, 0.1 rps; and spray-chamber temperature, 2 °C. The mercury was analyzed by atomic absorption spectrometry using an AMA-254 single-purpose atomic absorption spectrometer (Altec, Praha, Czech Republic).

### 2.8. Statistical Analysis

The data were analyzed using GraphPad Prism 8.3.0 (538) 2019 (GraphPad Software, Inc., San Diego, CA, USA). Data for total PPH, CAF, and AA were analyzed using two-way analyses of variance. The model included effects for coffee type, roast, and the coffee type × roast interaction. Individual differences were determined using Tukey’s multiple-comparison post-test and were considered to be significant at *p <* 0.05. Unpaired *t*-tests were applied to assess the differences in heavy-metal contents between the two types of green beans.

## 3. Results

### 3.1. Bioactive Compounds of the Coffee Beans

Quantitative analyses of the COL-GR, COL-LIGHT, and COL-DARK samples identified caffeoylquinic acids (CQA), feruloylquinic acids (FQA) and dicaffeoylquinic acids (diCQA), 5-caffeoylshikimic acid (5-CSKA), 3-caffeoylshikimic acid (3-CSKA) and 4-caffeoylshikimic acid (4-CSKA), 5-O-p-coumaroylquinic acid^1^, and 3-O-p-coumaroylquinic acid glucoside^2^ (Table 1).

The chromatogram peak numbers in Figure 1 represent the main PPH of the COL-GR, COL-LIGHT, and COL-DARK samples as numbered in Table 1.

Quantitative analyses of the NIC-GR, NIC-LIGHT, and NIC-DARK samples identified caffeoylquinic acids (CQA), feruloylquinic acids (FQA) and dicaffeoylquinic acids (diCQA), 5-caffeoylshikimic acid (5-CSKA), 3-caffeoylshikimic acid (3-CSKA), and 4-caffeoylshikimic acid (4-CSKA) (Table 2).

The peak numbers in Figure 2 represent the main PPH of the Nicaraguan samples as numbered in Table 2.

### 3.2. Effect of Coffee Type and Roasting on CGA Content

The effects of the type of coffee and roasting on CGA content are presented in Table 3. Roasting affected all CGA (*p <* 0.001). The strong similarity of NIC-LIGHT and NIC-DARK in UV chromatograms was observed. Coffee type significantly affected all contents, except for 3-O-CQA and 3-O-FQA. Coffee type and roasting interacted among the majority of CGA (*p <* 0.001).

### 3.3. Effects of Coffee Type and Roasting on the Contents of Total PPH, CAF, and AA

Coffee type and roasting affected the contents of total PPH (*p <* 0.001, Table 4). The content was highest in COL-GR. Roasting affected CAF content (*p <* 0.011), but the contents did not differ among the coffee types. AA content was affected by roasting (*p <* 0.001) and was lower in both COL-DARK and NIC-DARK than COL-LIGHT and NIC-LIGHT, respectively (*p <* 0.01).

### 3.4. Heavy Metal Contents in the Green Coffee Beans

The contents of heavy metals did not differ between COL-GR and NIC-GR, except for Cd and Ni (Table 5). Cd (*p <* 0.001) and Ni (*p <* 0.03) contents, which were significantly higher in COL-GR than NIC-GR.

## 4. Discussion

CGA content differed between the COL and NIC beans, perhaps because the PPH content of green coffee beans can vary among varieties, geographic location, and post-harvest processes before roasting. Climate conditions might also increase the exposure and vulnerability of coffee to pests and diseases and thus affect PPH content [23]. Analysis of the bioactive compounds of the COL and NIC samples indicated the presence of phenolic acids in three main CGA groups: CQA, FQA, and diCQA. The CGA content in the green beans agreed with the published total CGA contents of 4–9.2% for *C. arabica* and 7–14.4% for *Coffea robusta* [24,25]. Isomer 5-O-CQA is generally the most abundant CGA that forms beneficial bioactive compounds in green and roasted beans. The contents of the 5-O-CQA and 4-O-CQA isomers in our study were highest in the green beans for both coffees, but the content of the 5-O-CQA isomer was much higher in COL-GR than NIC-GR. It is well known that during coffee roasting, major changes occur in coffee bean composition that influence the antioxidant capacity of melanoidins and CGA content in a coffee brew [26]. Roasting conditions decreased the contents of the 5-O-CQA isomer by 61–70% for the COL coffees and by 49–58% for the NIC coffees, consistent with the decrease in the content of this isomer in roasted *C. robusta* (47–72%) [27]. The contents of the 3-O-CQA and 4-O-CQA isomers of the COL and NIC coffees increased approximately two-fold after light and darker roasting, also in agreement with other results [27]. However, UV chromatograms showed a similarity between NIC-LIGHT and NIC-DARK polyphenols. This can be probably ascribed to coffee processing. Fermented coffee beans are higher in PPH contents than unfermented coffee beans [28], which is also consonant with our results for green coffee samples. However, the total PPH contents of all Nicaraguan coffee samples, NIC-LIGHT and NIC-DARK, were not significantly different (Table 4). Therefore, we hypothesize that the natural processing of green coffee beans (i.e., Nicaraguan coffee) can probably lead to better protection of PPH during light and medium roasting (i.e., 170–220 °C) [12]. The 5-O-CQA isomer content, however, is the main marker of the quality of coffee beans and various other natural products [29]. Studies with different roasting conditions have reported correlations between the quality of coffee beverages and isomer 5-O-CQA content in commercial [30] and specialty coffees [5]. The contents of diCQA isomers were also high in both green beans, especially COL-GR, where the 3,5-diCQA isomer content was as high as 7.06 mg/g DM. The content of the 3,5-diCQA isomer in our green *C. arabica* beans was significantly higher than the other isomers of this group, but the contents of diCQA isomers differ little in green *C. robusta* beans [31]. Furthermore, CGA contents in green *C. arabica* beans differ between geographic locations in Kenya and Columbia [32,33].

Studies comparing total PPH contents in specialty coffees are limited. A recent study of a variety of Ethiopian specialty coffee beans roasted from 150 to 200 °C for 7–15 min, however, reported total PPH contents of 3.75, 1.85, 1.80, and 0.43% for green and light-, medium-, and dark-roasted beans, respectively [5]. Total PPH contents in our study were slightly higher in both coffees. The total PPH contents in the COL coffees were 5.98, 3.55, and 3.12% for the green and the light and darker roasted beans, respectively. The NIC coffees had slightly lower total PPH contents of 4.19, 3.28, and 2.99% for the green and the light and darker roasted beans, respectively. The losses of total PPH during roasting were 40–50% and 22–29% for the COL and NIC coffees, respectively. The losses of total PPH were generally higher from the COL beans, with 11–45% losses at 190–216 °C, than from the roasted *C. robusta* beans, with total PPH losses of <24% at 190 °C [27].

Differences in total PPH contents can also be influenced by the method of cultivation, the origin of the coffee, storage conditions, climate condition, pests, and diseases. PPH can be a natural defense mechanism of plants against pests and diseases, so organic coffee might have a higher content because of fewer pesticides used in farming [34]. Generally, organic crops have higher concentrations of antioxidants, lower concentrations of cadmium, and a lower incidence of pesticide residues across regions and production seasons [35]. Therefore, organic coffee beans also have a higher content of total PPH than do conventional beans [12]. The health benefits of PPH in commercially available coffees vary with processing conditions and degree of roasting [36], but the average loss of total PPH from green to dark-roasted coffee can be nearly 93% [30]. Specialty coffees have a similar trend of lower total PPH contents after roasting, but the average losses were substantially lower in our experiment and another study [5]. The differences in the content of total PPH in specialty coffee beverages also depend on the method of preparation (e.g., Hario V60, espresso, and pour-over) [37].

CAF content was similar in both the COL and NIC coffees but increased with roasting time. Lightly roasted coffees contain more CAF, which decreases with more intense roasting due to the release of CAF from the cell walls of the roasted beans and/or as the beans lose weight from the degradation of other organic substances [13,27]. Similarly, CAF content varies substantially depending on the origin of the coffee and the method of roasting and beverage preparation. CAF content in green *C. arabica* beans ranges from 0.8 to 1.4% [24] and can also vary substantially in roasted beans of different cultivars [38]. The CAF content in most *C. arabica* cultivars is >0.8%, consistent with the typical Ethiopian standard for roasted coffee [39]. In addition, the higher CGA contents and CAF, which were consistent with the higher values of total PPH and antioxidant capacity, were observed in the coffee silverskin of *C. robusta* as compared to *C. arabica* [40]. The CAF content in our study was 1.1% in the green beans and 1.2–1.3% in the light and darker-roasted beans. The CAF content in Ethiopian specialty coffees is 1.5–1.6% for green beans but decreases greatly during roasting [5]. These findings for specialty coffees are consistent with recent studies suggesting that the level of CAF decreases with longer roasting times and is highest in light- and medium-roasted commercial coffees [12,41].

AA contents were 192 and 277 µg/kg in the COL-DARK and NIC-DARK beans and 457 and 413 µg/kg in the COL-LIGHT and NIC-LIGHT beans, respectively. AA contents were generally lower than in street and industrially processed powdered coffees (346 ± 19 to 701 ± 38 μg/kg and 442 ± 14 to 906 ± 7 μg/kg, respectively) [42]. AA contents in roasted and instant coffee recommended by the European Commission, however, are 400 and 850 μg/kg, respectively [43]. The AA contents of the lightly roasted COL and NIC beans were slightly higher than the EU standards, but the darker roasted beans (COL-DARK and NIC-DARK) had lower AA contents, probably due to the “development time” during roasting, when the AA content peaks after the first crack and then begins to decrease. Minimizing AA formation during the processing of coffees is highly desirable, so more aspects should be considered, such as roasting temperature, time of the roast, type of roaster, the velocity of the roasting air, humidity, and the degradation of PPH. *C. robusta* beans roasted under optimal conditions (e.g., 203 °C, dry air, low velocity of roasting air) have a relatively low AA content (31.9–85.8 μg/kg) and a moderate level of degradation of PPH [27]. Different types of roaster (e.g., drum, fluidized bed, or traditional) produce different levels of AA, but drum roasters produce the best results at a medium degree of roasting (175 °C) [5]. However, the function of the type of coffee and origin of roasted specialty coffee varieties must be also taken into account when evaluating the cup quality [5].

In contrast to roasted coffee, limited studies have compared the contents of heavy metals in green beans, but their results are mostly consistent with our results [44,45,46]. The contents of some metals in green beans change after roasting depending on the degree of roasting, but the levels of heavy metals in coffee are within the recommended limits [47]. The Cd content in our COL coffees was higher (0.148 ± 0.0241 mg/kg) than in the majority of studies [44,45,46,47]. Cd content is rarely determined in coffee beans compared to other heavy metals because their contents are relatively low. The average Cd content (0.013 ± 0.008 mg/kg) in green *C. arabica* and *C. robusta* beans from different geographical regions [44] is similar to the content in the NIC-GR beans in our experiment. The reason for the strongly significant difference (*p <* 0.001) between the COL-GR and NIC-GR Cd contents remains unknown. Of the other heavy metals, only the Ni content differed significantly between the COL-GR and NIC-GR beans. The total contents of heavy metals, especially Cd, in nature are > 55-fold higher in South America than North America, with the main sources of pollution from rock weathering, fertilizers, pesticides, mining, and manufacturing [48]. These sources may be responsible for the significant differences in the Cd and Ni contents in the green beans in our experiment. Cd is nevertheless a contaminant that can accumulate in the environment, with a good correlation between soil and plants [49], but Cd intake estimated from plant-based beverages is low [50]. Interestingly, some studies strongly support the use of spent coffee grounds as an effective and economical adsorbent for the removal of cadmium, lead, and probably of other metal species from both industrial and drinking water [51,52].

Farmers only use sensorial analyses to evaluate the quality of specialty coffee. We therefore compared the bioactive substances and dangers that arose during the processing of coffee in our experiment to those of the coffees previously described. These substances may or may not occur in specialty coffee, even if processed properly, e.g., lightly roasted. Specialty coffee is considered to have the highest quality on the market, but the customer, roaster, and farmer do not know if it also contains more bioactive compounds. Methods other than sensorial analyses should therefore be considered for identifying substances in specialty coffee and evaluating its quality. This study should contribute to the development of a more analytical definition of quality in the future.

## 5. Conclusions

Total polyphenol contents were higher in COL-GR than the NIC-GR, but the Cd and Ni contents were higher in the COL than the NIC coffees. The darker roasts of both types of coffees had lower AA contents and similar CAF and total PPH contents as the light roasts for both types of coffees. Our study demonstrates that specialty coffees and the method of their production can preserve the content of bioactive compounds but also minimize harmful contaminants, which can benefit the health of consumers. Drinking healthy and traceable high-quality specialty coffees should be a goals in the future.

## Figures and Tables

**Figure 1 foods-10-01310-f001:**
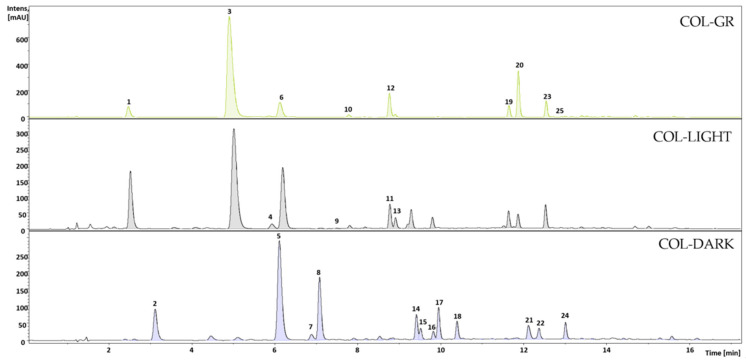
UV chromatograms at 320 nm for the 80% MeOH extracts of the green (COL-GR), lightly roasted (COL-LIGHT), and darker roasted (COL-DARK) Colombian coffee samples. Peak numbers represent the main compounds of polyphenols listed (No.) in Table 1.

**Figure 2 foods-10-01310-f002:**
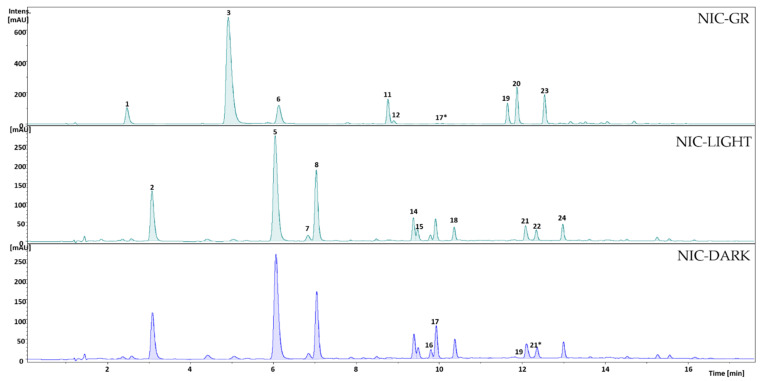
UV chromatograms at 320 nm of the 80% MeOH extracts of the green (NIC-GR), lightly roasted (NIC-LIGHT), and darker roasted (NIC-DARK) Nicaraguan coffee samples. Peak numbers represent the main polyphenols listed (No.) in Table 2.

**Table 1 foods-10-01310-t001:** Contents of main bioactive PPH (mg/g DM) in the Colombian coffee beans (COL-GR, COL-LIGHT, COL-DARK).

No.	RT (min)	UV (nm)	*m/z*[M-H]^−^	MS^2^ Main Ion	MS^2^ Fragments	Formula	Compound	COL-GR	COL-LIGHT	COL-DARK
1	2.5	215,325	353.0874	191.0548	179,161,135	C_16_H_18_O_9_	3-O-CQA	2.35 ± 0.08	5.05 ± 0.74	
2	3.1	215,325	353.0871	191.0556	179,161,135	C_16_H_18_O_9_	trans 3-O-CQA			2.92 ± 0.41
3	5.2	215,325	353.0881	191.0548	179,135,161	C_16_H_18_O_9_	5-O-CQA	36.3 ± 2.04	14.0 ± 1.02	
4	5.9	215,325	367.103	193.0493	134,149,173	C_17_H_20_O_9_	3-O-FQA	0.24 ± 0.02	0.55 ± 0.02	
5	6.1	215,325	353.0877	191.0553		C_16_H_18_O_9_	trans 5-O-CQA			10.8 ± 0.86
6	6.2	215,325	353.0877	191.0553	179,173	C_16_H_18_O_9_	4-O-CQA	3.93 ± 0.06	6.55 ± 0.21	
7	6.8	215,325	367.1033	193.0493	134,149,173	C_17_H_20_O_9_	trans 3-O-FQA			0.51 ± 0.02
8	7.1	215,325	353.0882	173.0439	179,191,134,155	C_16_H_18_O_9_	trans 4-O-CQA			5.05 ± 0.08
9	7.5	215,325	367.1034	161.0224	133	C_17_H_20_O_9_	4-O-FQA		0.05 ± 0.01	
10	7.8	215,325	337.0919	191.0545	173,163	C_16_H_18_O_8_	5-O-p-CoQA ^1^	0.43 ± 0.01		
11	8.8	215,325	335.0777	161.0228	135	C_16_H_16_O_9_	5-CSKA		1.60 ± 0.04	
12	8.9	215,325	367.1036	173.0438	193,134	C_17_H_20_O_9_	5-O-FQA	3.89 ± 0.09	0.72 ± 0.02	
13	9.3	215,325	335.0778	161.0227	135	C_16_H_16_O_9_	trans 4-CSKA		1.35 ± 0.07	
14	9.4	215,325	367.104	191.0548	173,155,134	C_17_H_20_O_9_	trans 5-O-FQA			1.53 ± 0.02
15	9.5	215,325	367.1044	173.0439	191,156,134	C_17_H_20_O_9_	trans 4-O-FQA			0.72 ± 0.02
16	9.8	215,325	335.0774	161.0224	179,135	C_16_H_16_O_8_	3-CSKA			0.45 ± 0.01
17	9.9	215,325	335.0777	161.0228	135	C_16_H_16_O_9_	5-CSKA			1.86 ± 0.07
18	10.4	215,325	335.0778	161.0227		C_16_H_16_O_9_	4-CSKA			1.06 ± 0.03
19	11.7	215,325	515.1199	353.0869	173,179,191,161	C_25_H_24_O_12_	3,4-DiCQA	0.05 ± 0.001	0.96 ± 0.07	
20	11.9	215,325	515.1195	191.0547	353,179	C_25_H_24_O_12_	4,5-DiCQA	1.69 ± 0.02	0.81 ± 0.02	
21	12.1	215,325	515.1196	353.0869	173,179,191,161	C_25_H_24_O_12_	trans 3,4-DiCQA			0.95 ± 0.01
22	12.4	215,325	515.1195	191.0547	353,179	C_25_H_24_O_12_	trans 4,5-DiCQA			0.77 ± 0.01
23	12.5	215,325	515.1204	353.0874	173,179,191	C_25_H_24_O_13_	3,5-DiCQA	7.06 ± 0.74	1.45 ± 0.02	
24	13	215,325	515.1197	353.0874	173,179,191	C_25_H_24_O_13_	trans 3,5-DiCQA			0.97 ± 0.01
25	13	215	499.1246	337.0928	163	C_25_H_24_O_11_	3-O-p-CoQA-Gluc ^2^	2.47 ± 0.04		

^1^ Coumaroylquinic acids; ^2^ Coumaroylquinic acid glucoside; DM, dry matter. Values are means ± SDs. SD, standard deviation.

**Table 2 foods-10-01310-t002:** Contents of main bioactive PPH (mg/g DM) in the Nicaraguan coffee beans (NIC-GR, NIC-LIGHT, NIC-DARK).

No.	RT (min)	UV (nm)	*m/z*[M-H]^−^	MS^2^ Main Ion	MS^2^ Fragments	Formula	Compound	NIC-GR	NIC-LIGHT	NIC-DARK
1	2.4	205,289	353.0874	191.0546	179,155,135	C_16_H_18_O_9_	3-O-CQA	2.22 ± 0.60		
2	3.1	215,325	353.0871	191.0556	179,161,135	C_16_H_18_O_9_	trans 3-O-CQA		4.40 ± 0.21	3.84 ± 0.07
3	4.9	215,325	353.0872	191.0556		C_16_H_18_O_9_	5-O-CQA	24.2 ± 2.11		
5	6.1	215,325	353.0877	191.0553		C_16_H_18_O_9_	trans 5-O-CQA		12.3 ± 1.22	10.1 ± 1.01
6	6.2	215,325	353.0884	173.044	179,191,135,155	C_16_H_18_O_9_	4-O-CQA	2.97 ± 0.08		
7	6.8	215,325	367.1033	193.0493	134,149,173	C_17_H_20_O_9_	trans 3-O-FQA		0.53 ± 0.04	0.40 ± 0.02
8	7.1	215,325	353.0882	173.0439	179,191,134,155	C_16_H_18_O_9_	trans 4-O-CQA		5.99 ± 0.65	4.67 ± 0.08
11	8.8	215,325	367.1044	191.0552	173	C_17_H_20_O_9_	5-O-FQA	2.58 ± 0.04		
12	8.9	215,325	367.1041	191.0548	173,155,134	C_17_H_20_O_9_	3-O-FQA	0.31 ± 0.01		
14	9.4	215,325	367.104	191.0548	173,155,134	C_17_H_20_O_9_	trans 5-O-FQA		1.53 ± 0.07	1.41 ± 0.06
15	9.5	215,325	367.1044	173.0439	191,156,134	C_17_H_20_O_9_	trans 4-O-FQA		0.71 ± 0.01	0.58 ± 0.02
16	9.8	215,325	335.0774	161.0224	179,135	C_16_H_16_O_8_	3-CSKA		0.39 ± 0.01	0.48 ± 0.01
17	9.9	215,325	335.0777	161.0228	135	C_16_H_16_O_9_	5-CSKA		1.57 ± 0.02	1.89 ± 0.04
17 *	9.9	215,325	367.1038	179.033	161,135	C_17_H_20_O_9_	4-O-FQA	0.05 ± 0.01		
18	10.4	215,325	335.0778	161.0227		C_16_H_16_O_9_	4-CSKA		0.84 ± 0.01	
19	11.7	215,325	515.1199	353.0869	173,179,191,161	C_25_H_24_O_12_	3,4-DiCQA	3.52 ± 0.06		
20	11.9	215,325	515.1195	191.0547	353,179	C_25_H_24_O_12_	4,5-DiCQA	1.90 ± 0.02		
21	12.1	215,325	515.1196	353.0869	173,179,191,161	C_25_H_24_O_12_	trans 3,4-DiCQA		1.05 ± 0.01	0.94 ± 0.02
21 *	12.2	215	349.0927	175.0382	160,193	C_17_H_18_O_8_	3-F-1,5-Quinolac			0.68 ± 0.01
22	12.4	215,325	515.1195	191.0547	353,179	C_25_H_24_O_12_	trans 4,5-DiCQA		0.66 ±0.02	0.66 ± 0.01
23	12.5	215,325	515.1204	353.0874	173,179,191	C_25_H_24_O_13_	3,5-DiCQA	2.68 ± 0.11		
24	13	215,325	515.1197	353.0874	173,179,191	C_25_H_24_O_13_	trans 3,5-DiCQA		1.00 ± 0.01	0.91 ± 0.01

Values are means ± SDs. SD- standard deviation, DM- dry matter, numbering in accordance with Table 1, * additional peaks compared to the COL samples.

**Table 3 foods-10-01310-t003:** Effects of coffee type and roasting on the contents of chlorogenic acids (mg/g DM).

C	R	CQA (Cis and Trans)			FQA (Cis and Trans)			diCQA (Cis and Trans)		
		3-O-CQA	5-O-CQA	4-O-CQA	3-O-FQA	5-O-FQA	4-O-FQA	3,4-diCQA	4,5-diCQA	3,5-diCQA
COL	GR	2.35 ± 0.08 ^a^	36.3 ± 2.04 ^b^	3.93 ± 0.06 ^b^	0.24 ± 0.02 ^a^	3.89 ± 0.09 ^d^	0.0 ± 0.00 ^a^	0.05 ± 0.01 ^a^	1.69 ± 0.02 ^d^	7.06 ± 0.74 ^d^
	LIGHT	5.05 ± 0.74 ^b^	14.0 ± 1.02 ^a^	6.55 ± 0.21 ^e^	0.55 ± 0.02 ^b^	0.72 ± 0.01 ^a^	0.05 ± 0.02 ^b^	0.96 ± 0.07 ^b^	0.81 ± 0.02 ^c^	1.45 ± 0.02 ^b^
	DARK	2.92 ± 0.41 ^b^	10.8 ± 0.86 ^a^	5.0 ± 0.09 ^c^	0.51 ± 0.02 ^b^	1.53 ± 0.02 ^b^	0.72 ± 0.02 ^c^	0.95 ± 0.01 ^b^	0.77 ± 0.01 ^c^	0.97 ± 0.01 ^a^
NIC	GR	2.22 ± 0.60 ^a^	24.2 ± 2.11 ^b^	2.97 ± 0.08 ^a^	0.31 ± 0.01 ^a^	2.58 ± 0.04 ^c^	0.05 ± 0.01 ^a^	3.52 ± 0.06 ^d^	1.90 ± 0.02 ^e^	2.68 ± 0.11 ^c^
	LIGHT	4.40 ± 0.21 ^b^	12.3 ± 1.22 ^a^	5.99 ± 0.65 ^d^	0.53 ± 0.04 ^b^	1.53 ± 0.07 ^b^	0.71 ± 0.01 ^c^	1.05 ± 0.01 ^c^	0.66 ± 0.02 ^b^	1.00 ± 0.01 ^a^
	DARK	3.84 ± 0.07 ^b^	10.1 ± 1.01 ^a^	4.67 ± 0.08 ^c^	0.40 ± 0.02 ^b^	1.41 ± 0.06 ^a^	0.58 ± 0.02 ^d^	0.94 ± 0.02 ^b^	0.66 ± 0.01 ^a^	0.91 ± 0.01 ^a^
Significance of effects										
Coffee type (C)		0.148	0.001	0.001	0.492	0.001	0.001	0.001	0.001	0.001
Roasting (R)		0.001	0.001	0.001	0.001	0.001	0.001	0.001	0.001	0.001
C × R		0.001	0.001	0.053	0.068	0.001	0.001	0.001	0.001	0.001

Values are means ± SDs. SD, standard deviation. Different letters within a column indicate significant differences at *p* < 0.05.

**Table 4 foods-10-01310-t004:** Effects of coffee type and roasting on the contents of total polyphenols, caffeine, and acrylamide.

Coffee	Roasting	Total Polyphenols (mg/g DM)	Caffeine (mg/kg)	Acrylamide (µg/kg)
Colombian	Green	59.8 ± 8.94 ^b^	11,104 ± 777	Nd
	Light	35.5 ± 3.64 ^a^	12,660 ± 886	457 ± 82.3 ^b^
	Darker	30.1 ± 4.56 ^a^	12,329 ± 863	192 ± 34.6 ^a^
Nicaraguan	Green	41.9 ± 6.44 ^a^	11,127 ± 779	Nd
	Light	32.8 ± 4.15 ^a^	12,822 ± 898	413 ± 74.3 ^b^
	Darker	30.0 ± 2.56 ^a^	12,868 ± 901	277 ± 49.9 ^a^
Significance of effects				
Coffee (C)		0.002	0.559	0.590
Roasting (R)		0.001	0.011	0.001
C × R		0.054	0.865	0.115

Values are means ± SDs. SD, standard deviation; Nd, not detected. Different letters within a column indicate significant differences at *p* < 0.05.

**Table 5 foods-10-01310-t005:** Contents of heavy metals (mg/kg, means ± SDs) in the green specialty coffee beans.

Heavy Metal	Colombian	Nicaraguan	*p*
Mercury	0.0007 ± 0.00003	0.0007 ± 0.00001	0.626
Copper	10.9 ± 2.33	14.1 ± 3.16	0.225
Lead	0.13 ± 0.02	0.10 ± 0.04	0.341
Chrome	0.06 ± 0.01	0.06 ± 0.01	0.124
Cadmium	0.15 ± 0.02	0.01 ± 0.01	0.001
Nickel	0.63 ± 0.07	0.48 ± 0.03	0.03
Aluminum	4.27 ± 0.53	3.28 ± 0.34	0.054

## Data Availability

Data available upon reasonable request to the corresponding author.

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
