# Peer review of "Heavy-Metal Contents and the Impact of Roasting on Polyphenols, Caffeine, and Acrylamide in Specialty Coffee Beans"

_foods, 2021, doi:10.3390/foods10061310_

Round 1

Reviewer 1 Report

Dear Authors

Thank you for your manuscript about specialty coffee beans. It is very good readable and understandable. However I have 5 important suggestions to improve your manuscript, as well as some minor suggestions, see below.

Important suggestions:

  • You describe nicely the green coffees you used, including the type of processing. Please mention already in the abstract and as well also when discussing your results, that the green coffees were processed totally different, namely the Nicaraguan samples are naturally processed, while the Colombian ones are fermented anaerobically!
  • Please include this difference in processing in the introduction and the discussion. Is there alreadyliterature about the influence of the type of processing with respect to your analyzed components like AA, CGA,...?
  • The effect of roast degree on the contents of AA, CAFand PPH have been studied already extensively. Please include more references.
  • Please describe how youdefine specialty coffee.
  • Discussion: last paragraph, lines 349-355: Is it copy paste from a different manuscript?It has no context here.

Further suggestions:

  • Table 1-7: Please include the standard deviations in the results.
  • Table 8 and 9: please reduce the numbers to the significant digits.
  • Figure 1/ 2: UV chromatograms of Colombian coffees were very similar for GR and LIGHT. For Nicaraguan, the UV chromatograms for LIGHT and DARK were very similar. OR IDENTICAL? They look the same. Please check. If they are not the same, please discuss the similarity of COL_GR and LIGHTversus the similarity of NIC_LIGHT and DARK.
  • Line 229-233: mention the different processing types here as well.It would be very interesting to know about the possible differences in CGA content etc for the very same coffee but different type of processing.
  • Line 249-252: Do you also have samples of the soil where the coffee grew? A comparison of heavy metal content of beans and soil would be interesting.
  • Line 249-252: Do you know what kind of fertilizer or pesticides the farmers use? This might have ahugeinfluence, no?
  • Discussion, line 256-258: PPH content is also drastically influenced by climatic conditions and pests and diseases.
  • Discussion, line 258ff: This is already well known,please include more references. It is for example nicelydiscussed in Smrke et al., Food Funct., 2013,4, 1082-1092, https://doi.org/10.1039/C3FO30377B.
  • Discussion lines 290 ff: again: influence of climatic conditions, pests and diseases. PPH can be a natural defence mechanism of plants against pests and diseases, so organic coffee might have a higher content because of less pesticides used in farming.
  • Discussionlines 325-330: This might alsobe a function of type of coffee, of origin, etc, no?
  • Discussionlines339-348: again, do you also have soil samples? DO you know the type/amount of fertilizers or pesticides the farmers used?
  • Why did you not measure the content of heavy metals in the roasted samples?

Author Response

Revision Note

 REVIEWER 1 (R1):

 We appreciate your valuable comments and effort in revising our manuscript. Thank you very much and we hope our corrections and additions have improved the manuscript sufficiently for publication. The responses to comments were highlighted by red color in the revised manuscript and identified below in the Revision note.

 Revision Note

Important suggestions:

R1: You describe nicely the green coffees you used, including the type of processing. Please mention already in the abstract and as well also when discussing your results, that the green coffees were processed totally different, namely the Nicaraguan samples are naturally processed, while the Colombian ones are fermented anaerobically!

AUTHORS: In the Abstract and the Discussion the coffee processing was involved according to R1 comments.  Abstract Line 19-20: “Samples of NIC were naturally processed and COL has fermented anaerobically.” Discussion: Line 272-274: “The factor affecting PPH content is also coffee processing because samples of NIC were naturally processed and COL has fermented anaerobically.”

R1: Please include this difference in processing in the introduction and the discussion. Is there already literature about the influence of the type of processing with respect to your analyzed components like AA, CGA,...?

AUTHORS: The influence of the type of coffee processing on bioactive compounds is not available in the literature. The new innovative types of coffee processing (anaerobic fermentation, carbonic macerations, thermal shock, etc.) to maintain the quality of coffee has only been used recently by farmers and therefore our contribution is unique. Currently, our research will be focused on this topic also in the future (i.e., one coffee from one farm processed by different methods of processing).

 R1: The effect of roast degree on the contents of AA, CAF and PPH have been studied already extensively. Please include more references.

AUTHORS: According to the R1 comments we included the sentence with 3 references in the Introduction of the manuscript: Line 68-69: “The effect of roast degree on the contents of AA, CAF, and polyphenols (PPH) has been studied already extensively [17-19].”

[17] Vignoli, J.A.; Bassoli, D.G.; Benassi, M.T. Antioxidant activity, polyphenols, caffeine and melanoidins in soluble coffee: The influence of processing conditions and raw material. Food Chem. 2011, 124, 863–868. https://doi.org/10.1016/j.foodchem.2010.07.008 

[18] Vignoli, J.A.; Viegas, M.C.; Bassoli, D.G.; Benassi, M.T. Roasting process affects differently the bioactive compounds and the antioxidant activity of Arabica and Robusta coffees. Food Res. Int. 2014, 61, 279–285. https://doi.org/10.1016/j.foodres.2013.06.006  

[19] Dybkowska, E.; Sadowska, A.; Rakowska, R.; Dębowska, M.; Świderski, F.; Świąder, K. Assessing polyphenols content and antioxidant activity in coffee beans according to origin and the degree of roasting. 2017, Rocz. Państw. Zakł. Hig. 68, 347–353. PMID: 29265388   

R1: Please describe how you define specialty coffee.

AUTHORS: In my opinion specialty coffee could be defined as a term for coffee that has a standardized whole coffee process cycle of production from choosing coffee plantation criteria to coffee brew which pass preliminary grading and cupping tests and only then can be serving to the client. One of the most important criteria is to achieve a cupping score of 80 points or above on the 100 point scale. Farmers are using the new types of coffee processing (anaerobic fermentation, carbonic macerations, thermal shock, etc.) to maintain the quality of coffee. In our contribution, the Colombian coffee was processed by anaerobic fermentation which is a brand new fermentation technique and it has not previously been investigated what effect fermentation has on the occurrence of bioactive compounds. (It was added in the Abstract of the manuscript). The production of Coffea Arabica is decreasing every year, farmers are using the new introgressive species hybrids to maintain the quality and production for new decades. In our case, the coffee from Nicaragua is variety Sarchimor- a natural cross between C. arabica and C. canephora. It is tolerant of coffee berry diseases and has the potential of quality coffee. And the most important thing that it scores more than 80 points on the 100 point scale which we can not say about any other C. canephora coffee. In our point of view, specialty coffee is a new and innovative way of the coffee industry, where new, previously unused processes, are used.

Therefore, we wanted to compare the bioactive substances and dangers that arise in the processing of coffee, compared to the coffees that were described in the past. Whether the specialty coffee, even if processed properly and correctly, light roasted, has or does not have any other occurrence of these substances. Specialty coffee is considered to be the best in quality on the market ( it is just 5% of the whole market) but the customer, roaster, and farmer can not say that if it also contains more bioactive compounds. so not only sensory analysis should be considered as a major in the evaluation of specialty coffee but also other methods for the determination of substances in specialty coffee to clarify its quality.

Line 45-48: We added in the text of Introduction: “One of the most important criteria is to achieve a cupping score of 80 points or above on the 100 point scale [4]. Farmers who process specialty coffee are using the new types of coffee processing (anaerobic fermentation, carbonic macerations, thermal shock, etc.) to maintain the quality of coffee.”

R1: Discussion: last paragraph, lines 349-355: Is it copy paste from a different manuscript? It has no context here.

AUTHORS: The last paragraph was removed from the manuscript.

Further suggestions:

R1: Table 1-7: Please include the standard deviations in the results.

AUTHORS: The standard deviations were included in Tables 1-7.

R1: Table 8 and 9: please reduce the numbers to the significant digits.

AUTHORS: The numbers in Tables 8 and 9 were reduced to significant digits.

R1: Figure 1/ 2: UV chromatograms of Colombian coffees were very similar for GR and LIGHT. For Nicaraguan, the UV chromatograms for LIGHT and DARK were very similar. OR IDENTICAL? They look the same. Please check. If they are not the same, please discuss the similarity of COL_GR and LIGHT versus the similarity of NIC_LIGHT and DARK.

AUTHORS: We made three separate weights from each coffee bag, separately for the Nicaraguan and separately for the Colombian. We include the “raw” UV chromatograms in the manuscript (in new Figure 3) for a better representation of differences and similarities. The similarity of COL-GR and COL-LIGHT was not observed. However, the similarity of NIC-LIGHT and NIC-DARK was found. It is discussed in the Discussion.  

R1: Line 229-233: mention the different processing types here as well. It would be very interesting to know about the possible differences in CGA content etc for the very same coffee but different type of processing.

AUTHORS: Line 243-244 in Results: We include the new sentence in the manuscript according to R1 comments: “The strong similarity of NIC-LIGHT and NIC-DARK in UV chromatograms was observed (Figure 3).”

The similarity was discussed in the Discussion in Line 289-296:  “However, UV chromatograms showed the similarity between NIC-LIGHT and NIC-DARK polyphenols. It can be probably ascribed to coffee processing. Fermented coffee beans are higher in PPH contents than unfermented coffee beans [28], which is also consonant with our results for green coffee samples. However, the total PPH contents of all Nicaraguan coffee samples, COL-LIGHT and COL-DARK were not significantly different (Table 8). It can be explained that natural processing of green coffee beans (i.e., Nicaraguan coffee) can probably lead to better protection of PPH during roasting.”

New reference was involved in the manuscript:

[28] Haile, M.; Kang, W.H. Antioxidant Activity, Total Polyphenol, Flavonoid and Tannin Contents of Fermented Green Coffee Beans with Selected Yeasts. Fermentation. 2019, 5, 29. https://doi.org/10.3390/fermentation5010029

R1: Line 249-252: Do you also have samples of the soil where the coffee grew? A comparison of heavy metal content of beans and soil would be interesting.

AUTHORS: We completely agree with the Reviewer, that a comparison of the heavy metal content of beans and soil would be interesting. Unfortunately, we had not the soil from this area.

R1: Line 249-252: Do you know what kind of fertilizer or pesticides the farmers use? This might have a huge influence, no?

AUTHORS: We completely agree with the Reviewer, that kind of fertilizer or pesticides might have a huge influence. Unfortunately, we had not this information. The analysis of pesticide contamination will be a part of our research in the next experiments.

R1: Discussion, line 256-258: PPH content is also drastically influenced by climatic conditions and pests and diseases.

AUTHORS: Line 274-275: According to R1 comments the new sentence with reference was added in the text of the manuscript: “Climate condition might also increase exposure and vulnerability of coffee to pests and diseases and thus affect PPH content [23].“

[23] Pham, Y., Reardon-Smith, K., Mushtaq, S. et al. The impact of climate change and variability on coffee production: a systematic review. Climatic Change 156, 609–630 (2019). https://doi.org/10.1007/s10584-019-02538-y).

R1: Discussion, line 258ff: This is already well known, please include more references. It is for example nicely discussed in Smrke et al., Food Funct., 2013,4, 1082-1092, https://doi.org/10.1039/C3FO30377B.

AUTHORS: Line 282-284: According to R1 comments the new sentence with reference was added in the text of the manuscript: “It is well known that during coffee roasting, major changes occur in coffee bean composition that influences the antioxidant capacity of melanoidins and CGA content in a coffee brew [26]”

[26] Smrke S, Opitz SE, Vovk I, Yeretzian C. How does roasting affect the antioxidants of a coffee brew? Exploring the antioxidant capacity of coffee via on-line antioxidant assays coupled with size exclusion chromatography. Food Funct. 2013 4(7):1082-1092. doi: 10.1039/c3fo30377b.

R1: Discussion lines 290 ff: again: influence of climatic conditions, pests and diseases. PPH can be a natural defence mechanism of plants against pests and diseases, so organic coffee might have a higher content because of less pesticides used in farming.

AUTHORS: Line 317-324: The new sentences with 2 new references were added in the Discussion: “Differences in total PPH contents can also be influenced by the method of cultivation, the origin of the coffee, storage conditions, climate condition, pests, and diseases. PPH can be a natural defense mechanism of plants against pests and diseases, so organic coffee might have a higher content because of fewer pesticides used in farming [34]. Generally, organic crops, have higher concentrations of antioxidants, lower concentrations of cadmium, and a lower incidence of pesticide residues across regions and production seasons [35]. Therefore, the organic coffee beans also have a higher content of total PPH than do conventional beans [12].”

[34] Veberic, R. The Impact of Production Technology on Plant Phenolics. Horticulturae 2016, 2, 8. https://doi.org/10.3390/horticulturae2030008

[35] Barański, M.; Srednicka-Tober, D.; Volakakis, N.; Seal, C.; Sanderson, R.; Stewart, G. B.; Benbrook, C.; Biavati, B.; Markellou, E.; Giotis, C.; Gromadzka-Ostrowska, J.; Rembiałkowska, E.; Skwarło-Sońta, K.; Tahvonen, R.; Janovská, D.; Niggli, U.; Nicot, P.; Leifert, C. Higher antioxidant and lower cadmium concentrations and lower incidence of pesticide residues in organically grown crops: a systematic literature review and meta-analyses. Br. J. Nutr. 2014, 112(5), 794–811. https://doi.org/10.1017/S0007114514001366 

R1: Discussion lines 325-330: This might also be a function of type of coffee, of origin, etc, no?

AUTHORS: Line 363-365: According to R1 comments we involved the missing sentence in Discussion: “However, the function of the type of coffee and origin of roasted specialty coffee varieties must be also taken into account when evaluated the cup quality [5].”    

R1: Discussion lines 339-348: again, do you also have soil samples? DO you know the type/amount of fertilizers or pesticides the farmers used?

AUTHORS: Unfortunately, we had not to soil from this area. We have not information about the type/amount of fertilizers or pesticides used by farmers.

R1: Why did you not measure the content of heavy metals in the roasted samples?

AUTHORS: For us, it was a pilot study, heavy metals were measured only in green coffee, because according to the available literature, the amount of heavy metals does not decrease by roasting significantly, so it is important to measure them in green coffee.

Reviewer 2 Report

This article reports on a study that investigated the effect of roasting on the contents in phenolic compounds, acrylamide and caffeine in specialty coffee beans from two different origins Colombia and Nicaragua. Further the authors determined the presence and amounts of heavy metals in green coffee samples. The effects of roasting and origin on coffee phenolics, caffeine and acrylamide have been widely studied for decades and the novelty of the present study highly questionable. Authors claim that contents of these compounds have been poorly studied in specialty coffees. The differential point of specialty coffee is just a standardized process of production which guarantees a certain level of quality which does not justify the novelty of this work. This fact is supported by the results that are in accordance with the vast amount of research articles on effect of roasting on coffee bioactives.

Author Response

REVIEWER 2 (R2):

We appreciate your valuable comments and effort in revising our manuscript. The responses to all Reviewer' comments were highlighted by red color in the revised manuscript.

Revision Note

R2: This article reports on a study that investigated the effect of roasting on the contents in phenolic compounds, acrylamide and caffeine in specialty coffee beans from two different origins Colombia and Nicaragua. Further the authors determined the presence and amounts of heavy metals in green coffee samples. The effects of roasting and origin on coffee phenolics, caffeine and acrylamide have been widely studied for decades and the novelty of the present study highly questionable. Authors claim that contents of these compounds have been poorly studied in specialty coffees. The differential point of specialty coffee is just a standardized process of production which guarantees a certain level of quality which does not justify the novelty of this work. This fact is supported by the results that are in accordance with the vast amount of research articles on effect of roasting on coffee bioactives.

AUTHORS:

Specialty coffee could be defined as a term for coffee that has a standardized whole coffee process cycle of production from choosing coffee plantation criteria to coffee brew which pass preliminary grading and cupping tests and only then can be serving to the client. One of the most important criteria is to achieve a cupping score of 80 points or above on the 100 point scale. (added in the Introduction).

Farmers are using the new types of coffee processing (anaerobic fermentation, carbonic macerations, thermal shock, etc.) to maintain the quality of coffee. (added in the Introduction). In our contribution, the Colombian coffee was processed by anaerobic fermentation which is a brand new fermentation technique and it has not previously been investigated what effect fermentation has on the occurrence of bioactive compounds. (added in the Abstract of the manuscript).

The production of Coffea Arabica is decreasing every year, farmers are using the new introgressive species hybrids to maintain the quality and production for new decades. In our case, the coffee from Nicaragua is variety Sarchimor- a natural cross between C. arabica and C. canephora. It is tolerant of coffee berry diseases and has the potential of quality coffee. And the most important thing that it scores more than 80 points on the 100 point scale which we can not say about any other C. canephora coffee. In our point of view, specialty coffee is a new and innovative way of the coffee industry, where new, previously unused processes, are used.

Therefore, we wanted to compare the bioactive substances and dangers that arise in the processing of coffee, compared to the coffees that were described in the past. Whether the specialty coffee, even if processed properly and correctly, light roasted, has or does not have any other occurrence of these substances. Specialty coffee is considered to be the best in quality on the market ( it is just 5% of the whole market) but the customer, roaster, and farmer can not say that if it also contains more bioactive compounds. so not only sensory analysis should be considered as a major in the evaluation of specialty coffee but also other methods for the determination of substances in specialty coffee to clarify its quality.

Reviewer 3 Report

This is an interesting article that, in my opinion, is either well written.

Unfortunately the main issues are regarding a better discussion of results and the novelty regarding the same.

There are some aspects that are not properly addressed. For example, just few articles that are not even discussed and are of importance for this topic:
DOI: 10.3303/CET1647050
DOI: 10.1016/j.foodres.2017.05.017

I hope authors revise and stress better the importance in comparison with previous literature.
I kindly ask even for a better English language.

Before the above mentioned improvements, the overall assessment is: Major.

Regards.

Author Response

REVIEWER 3 (R3):

We appreciate your valuable comments and effort in revising our manuscript. Thank you very much and we hope our corrections and additions have improved the manuscript sufficiently for publication. The responses to all Reviewer' comments were highlighted by red color in the revised manuscript.

Revision Note

R3: Unfortunately, the main issues are regarding a better discussion of results and the novelty regarding the same.

AUTHORS: We try to highlight the novelty of the study according to R3 comments. Specialty coffee could be defined as a term for coffee that has a standardized whole coffee process cycle of production from choosing coffee plantation criteria to coffee brew which pass preliminary grading and cupping tests and only then can be serving to the client. One of the most important criteria is to achieve a cupping score of 80 points or above on the 100 point scale. Farmers are using the new types of coffee processing (anaerobic fermentation, carbonic macerations, thermal shock, etc.) to maintain the quality of coffee. (added in the Introduction). In our contribution, the Colombian coffee was processed by anaerobic fermentation which is a brand new fermentation technique and it has not previously been investigated what effect fermentation has on the occurrence of bioactive compounds. (added in the Abstract). The production of Coffea Arabica is decreasing every year, farmers are using the new introgressive species hybrids to maintain the quality and production for new decades. In our case, the coffee from Nicaragua is variety Sarchimor- a natural cross between C. arabica and C. canephora. It is tolerant of coffee berry diseases and has the potential of quality coffee. And the most important thing that it scores more than 80 points on the 100 point scale which we can not say about any other C. canephora coffee. In our point of view, specialty coffee is a new and innovative way of the coffee industry, where new, previously unused processes, are used. Therefore, we wanted to compare the bioactive substances and dangers that arise in the processing of coffee, compared to the coffees that were described in the past. Whether the specialty coffee, even if processed properly and correctly, light roasted, has or does not have any other occurrence of these substances. Specialty coffee is considered to be the best in quality on the market ( it is just 5% of the whole market) but the customer, roaster, and farmer can not say that if it also contains more bioactive compounds. so not only sensory analysis should be considered as a major in the evaluation of specialty coffee but also other methods for the determination of substances in specialty coffee to clarify its quality.

R3: There are some aspects that are not properly addressed. For example, just few articles that are not even discussed and are of importance for this topic:
DOI: 10.3303/CET1647050
DOI: 10.1016/j.foodres.2017.05.017

AUTHORS: Thank you very much for your comment. The mentioned articles were discussed and 3 new references were involved in the revised manuscript in Discussion.

Line 339-341: “In addition, the higher CGA contents and CAF which were consistent with the higher values of total PPH and antioxidant capacity was observed in the coffee silverskin of C. robusta as compared to C. arabica [41].”

Line 383-386: “Interestingly, some studies strongly support the use of spent coffee grounds as an effective and economical adsorbent for the removal of cadmium, lead, and probably of other metal species from both industrial and drinking water [52,53].”

[41] Panusa A, Petrucci R, Lavecchia R, Zuorro A. UHPLC-PDA-ESI-TOF/MS metabolic profiling and antioxidant capacity of arabica and robusta coffee silverskin: Antioxidants vs phytotoxins. Food Res Int. 2017, 99:155-165. doi: 10.1016/j.foodres.2017.05.017.

[52] Lavecchia R., Medici F., Patterer S., Zuorro A., 2016, Lead removal from water by adsorption on spent coffee grounds, Chemical Engineering Transactions, 47, 295-300. doi: 10.3303/CET1647050

[53]Patterer S., Bavasso I., Sambeth J., Medici F., 2017, Cadmium removal from acqueous solution by adsorption on spent coffee grounds, Chemical Engineering Transactions, 60, 157-162 DOI: 10.3303/CET1760027

R3: I kindly ask even for a better English language.

AUTHORS: The English have been revised throughout the whole manuscript by a native English language editor, Dr. William Blackhall. https://www.globalbiologicalediting.com/index.html

Round 2

Reviewer 1 Report

Dear Authors

Thank you for your corrections and answers. I still have some minor suggestions / comments:

Discussion:

Lines 272-273: «The factor affecting PPH content is also coffee processing because samples of NIC were naturally processed and COL has fermented anaerobically.”               You cannot take this conclusion, you have two different coffees from different countries etc. If you would compare the very same coffee from the very same farm and everything the same except the processing, then you can judge whether there the processing has an influence on the PPH content or not.

Lines 293-294: “However, the total PPH contents of all Nicaraguan coffee samples, COL-LIGHT and COL-DARK were not significantly different (Table 8).”           Nicaragua or Colombia?

Lines 294-295: “It can be explained that natural processing of green coffee beans (i.e., Nicaraguan coffee) can probably lead to better protection of PPH during roasting.”               How could you explain a better thermal stability of these molecules based on natural processing? Did other authors see this as well for naturally processed coffee?

difference in processing 

"AUTHORS: The influence of the type of coffee processing on bioactive compounds is not available in the literature.  … … Fermented coffee beans are higher in PPH contents than unfermented coffee beans [28],”

So there is literature data on that, see also for example

https://doi.org/10.3389/fmicb.2019.02621        Influence of Various Processing Parameters on the Microbial Community Dynamics, Metabolomic Profiles, and Cup Quality During Wet Coffee Processing

https://doi.org/10.1080/10408398.2015.1067759      Microbial ecology and starter culture technology in coffee processing

DOI: 10.1128/AEM.02635-18          Following Coffee Production from Cherries to Cup: Microbiological and Metabolomic Analysis of Wet Processing of Coffea arabica

Author Response

Dear Reviewer,

Thank you very much and we hope our corrections have improved the manuscript sufficiently. The responses to comments were highlighted by red color in the revised manuscript.

REVIEWER 1 (R1): 

Revision Note

Discussion:

R1: Lines 272-273: «The factor affecting PPH content is also coffee processing because samples of NIC were naturally processed and COL has fermented anaerobically.”               You cannot take this conclusion, you have two different coffees from different countries etc. If you would compare the very same coffee from the very same farm and everything the same except the processing, then you can judge whether there the processing has an influence on the PPH content or not.

AUTHORS: Thank you for your comment. The incorrect sentence was removed from the text of the manuscript.

R1: Lines 293-294: “However, the total PPH contents of all Nicaraguan coffee samples, COL-LIGHT and COL-DARK were not significantly different (Table 8).”  Nicaragua or Colombia?

AUTHORS: Line 288-289: Yes, correct is Nicaragua. We have changed “NIC-LIGHT and NIC-DARK” in the text of the manuscript.

R1: Lines 294-295: “It can be explained that natural processing of green coffee beans (i.e., Nicaraguan coffee) can probably lead to better protection of PPH during roasting.”               How could you explain a better thermal stability of these molecules based on natural processing? Did other authors see this as well for naturally processed coffee?

 AUTHORS: Lines 289-291: According to your valuable comments, we changed the sentence on: “Therefore, we hypothesize that natural processing of green coffee beans (i.e., Nicaraguan coffee) can probably lead to better protection of PPH during light and medium roasting (i.e., 170-220 °C) [12].”

difference in processing 

R1: "AUTHORS: The influence of the type of coffee processing on bioactive compounds is not available in the literature.  … … Fermented coffee beans are higher in PPH contents than unfermented coffee beans [28],”

So there is literature data on that, see also for example

https://doi.org/10.3389/fmicb.2019.02621        Influence of Various Processing Parameters on the Microbial Community Dynamics, Metabolomic Profiles, and Cup Quality During Wet Coffee Processing

https://doi.org/10.1080/10408398.2015.1067759      Microbial ecology and starter culture technology in coffee processing

DOI: 10.1128/AEM.02635-18          Following Coffee Production from Cherries to Cup: Microbiological and Metabolomic Analysis of Wet Processing of Coffea arabica

AUTHORS: Thank you very much for your comment. We misinterpreted the sentence that this theme is not available in the literature. Few types of research talk about this topic. However, it would be interesting to compare it with new coffee processing (anaerobic fermentation, carbonic maceration). Fermentation is a natural way and it occurs in every coffee processing method (natural, honey, washed) but anaerobic fermentation or carbonic maceration is created by the synthetic way(creating an anaerobic environment in barrels or vessels). 

Reviewer 3 Report

Authors improved carefully the manuscript.

Author Response

Dear Reviewer,

We appreciate your valuable comments and effort in revising our manuscript. Thank you very much and we hope our corrections have improved the manuscript sufficiently for publication. The responses to the Reviewer's comments were highlighted by red color in the revised manuscript.